# Genetic programming of macrophages generates an in vitro model for the human erythroid island niche

Martha Lopez-Yrigoyen [1], Cheng-Tao Yang[1,5], Antonella Fidanza[1], Luca Cassetta [2], A. Helen Taylor[1], Angela McCahill[3], Erica Sellink[4], Marieke von Lindern[4], Emile van den Akker[4], Joanne C. Mountford [3], Jeffrey W. Pollard [2] & Lesley M. Forrester [1]

Red blood cells mature within the erythroblastic island (EI) niche that consists of specialized macrophages surrounded by differentiating erythroblasts. Here we establish an in vitro system to model the human EI niche using macrophages that are derived from human induced pluripotent stem cells (iPSCs), and are also genetically programmed to an EI-like phenotype by inducible activation of the transcription factor, KLF1. These EI-like macrophages increase the production of mature, enucleated erythroid cells from umbilical cord blood derived CD34$^+$ haematopoietic progenitor cells and iPSCs; this enhanced production is partially retained even when the contact between progenitor cells and macrophages is inhibited, suggesting that KLF1-induced secreted proteins may be involved in this enhancement. Lastly, we find that the addition of three secreted factors, ANGPTL7, IL-33 and SERPINB2, significantly enhances the production of mature enucleated red blood cells. Our study thus contributes to the ultimate goal of replacing blood transfusion with a manufactured product.

[1] MRC Centre for Regenerative Medicine, University of Edinburgh, 5 Little France Drive, Edinburgh EH16 4UU, UK. [2] MRC Centre for Reproductive Health, University of Edinburgh, Queens Medical Research Institute, 47 Little France Crescent, Edinburgh EH16 4TJ, UK. [3] Scottish National Blood Transfusion Service, The Jack Copland Centre, 52 Research Avenue North, Heriot-Watt Research Park, Edinburgh EH14 4BE, UK. [4] Sanquin Research, Department of Hematopoiesis, The Netherlands and Landsteiner Laboratory, Academic University Medical Cente, University of Amsterdam Amsterdam, The Netherlands. [5] Present address: Adaptimmune, 60 Jubilee Avenue, Milton Park, Abingdon, Oxfordshire OX14 4RX, UK. Correspondence and requests for materials should be addressed to L.M.F. (email: L.Forrester@ed.ac.uk)

Macrophages are key players within the innate immune system, in the regulation of developmental processes and in adult tissue homoeostasis, remodelling and repair[1,2]. The vast range of macrophage functions is reflected in their phenotypic heterogeneity and plasticity[3].

Macrophages associated with the erythroblastic island (EI) niche provide an environment throughout the stages of red blood cell (RBC) proliferation and maturation in vivo and engulf free nuclei as they are extruded from the cell[4]. The molecular interactions between the EI macrophage and developing erythroid cells are poorly understood because the human EI niche is inaccessible and no appropriate culture models exist. This has hampered the identification of factors that could be used to diagnose and treat anaemia and/or in the production of RBCs in vitro from renewable sources for cell therapy. This is becoming increasingly important because, although blood transfusion remains the most prominent means of treating chronic haematological disorders and trauma, it faces serious problems with donor supply, cell quality, infection transmission and immune incompatibility[5,6]. Attempts have been made to produce RBCs in vitro from different starting cell populations including CD34+ haematopoietic progenitor cells (HPCs), pluripotent stem cells (PSCs) and more recently, immortalized erythroid progenitor cells but production is relatively inefficient and final steps of RBC maturation are variable[7–12].

In the murine system it is known that the macrophage–erythroblast interaction provides both positive and negative regulators of cell differentiation and development throughout the stages of erythroid proliferation and maturation[4]. We reasoned that the production of an in vitro model for the human EI niche in vitro would identify and characterize factors associated with RBC production and maturation that could be used to improve their production from renewable sources. The first hurdle in this process was to generate a population of macrophages that had a phenotype comparable to those of the EI niche. Human monocyte-derived macrophages can promote primary erythroblast proliferation and survival but differing

effects on maturation and enucleation have been reported[13,14]. Discrepancies could reflect the source and heterogeneous phenotype of the macrophage cell populations that were used and culture conditions[15]. Furthermore, as tissue resident macrophages are thought to have a distinct developmental origin, primary monocyte-derived macrophages might not accurately reflect the EI niche[16–19]. Macrophages derived from PSCs in vitro have been reputed to be more akin to tissue resident macrophages so we reasoned that they might provide a renewable source of cells to test factors that have been implicated with the EI niche[17,18].

We previously demonstrated that activation of the transcription factor KLF1 enhanced the maturation of iPSC-derived erythroid cells but this effect was only observed at a time point when the differentiating culture consisted of a heterogeneous mixture of haematopoietic cells[20]. As an extrinsic role of KLF1 within the murine erythroid island (EI) niche had been reported[21,22], we hypothesized that the effect of KLF1 activation in differentiating iPSCs could also be mediated by its action in macrophages that might be acting as support cells in this context. To test this hypothesis, we generated a pure population of macrophages from the iPSC line carrying an inducible *KLF1* transgene (iKLF1.2)[20].

Here we demonstrate that KLF1 activation is able to programme iPSC-derived macrophages into an EI-like phenotype as assessed by their marker expression and their increased phagocytic activity. Our data show that EI-niche-like macrophages enhance the production of functional, mature, enucleated RBCs in vitro, and also identify three secreted factors associated with this mechanism of action.

## Results

**IPSC-DMs express low levels of *KLF1*.** To address whether induced pluripotent stem cell-derived macrophages (iPSC-DMs) had a phenotype comparable to macrophages associated with the EI niche, we assessed their expression of genes encoding the transcription factors, *MAF* and *KLF1* (Fig. 1a)[22]. C-MAF was expressed at a significantly higher level in iPSC-DMs compared to

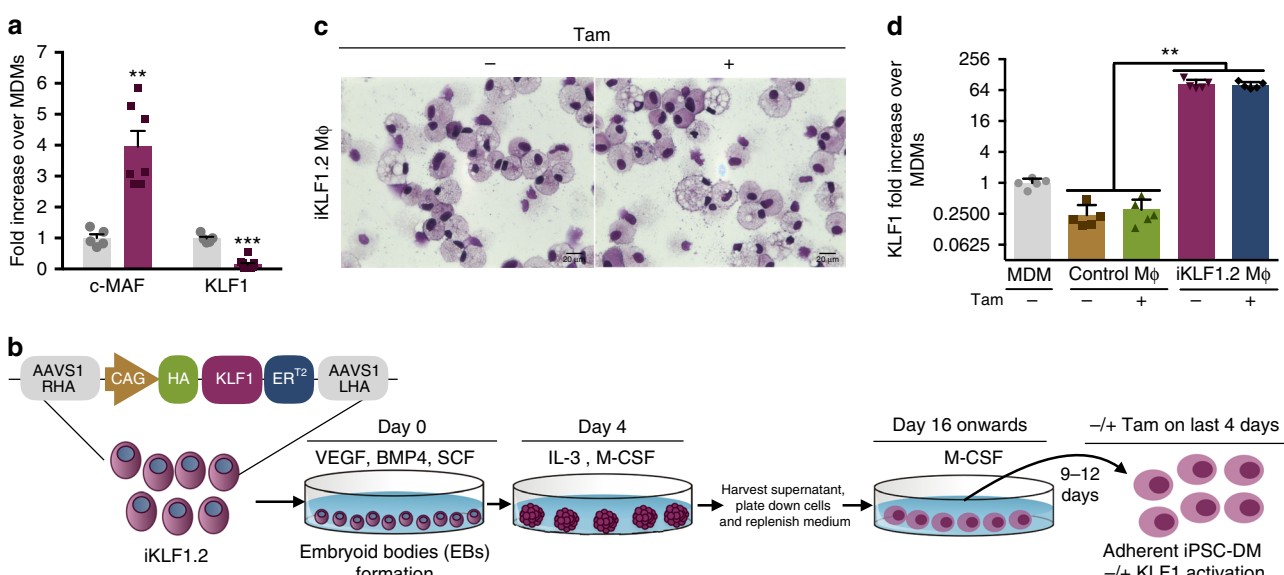

**Fig. 1** AAVS1-targeted *KLF1* transgene is expressed in iPSC-DMs. **a** Expression of 'EI' macrophage related transcription factors (*c-MAF* and *KLF1*) in monocyte-derived macrophages (MDMs) (grey bar) and iPSC-derived macrophages (purple bar) by qRT-PCR analyses (*n* = 5, Mann Whitney test). **b** Differentiation protocol used to generate macrophages from an iPSC line (iKLF1.2) carrying the AAVS1-targeted KLF1-ER[T2] transgene. **c** Kwik-Diff-stained cytospin preparations of iKLF1.2-derived macrophages (iKLF1.2 MΦ) (scale bar, 20 μm). **d** Real-time PCR analysis of *KLF1* in MDMs , control iPSC-DMs (control MΦ) and iKLF1.2-DMs (iKLF1.2 MΦ) (*n* = 4 biologically independent samples) Kruskal–Wallis test with Dunn's post-test). *\*p* < 0.05, *\*\*p* < 0.01, *\*\*\*p* < 0.001, *\*\*\*\*p* < 0.0001

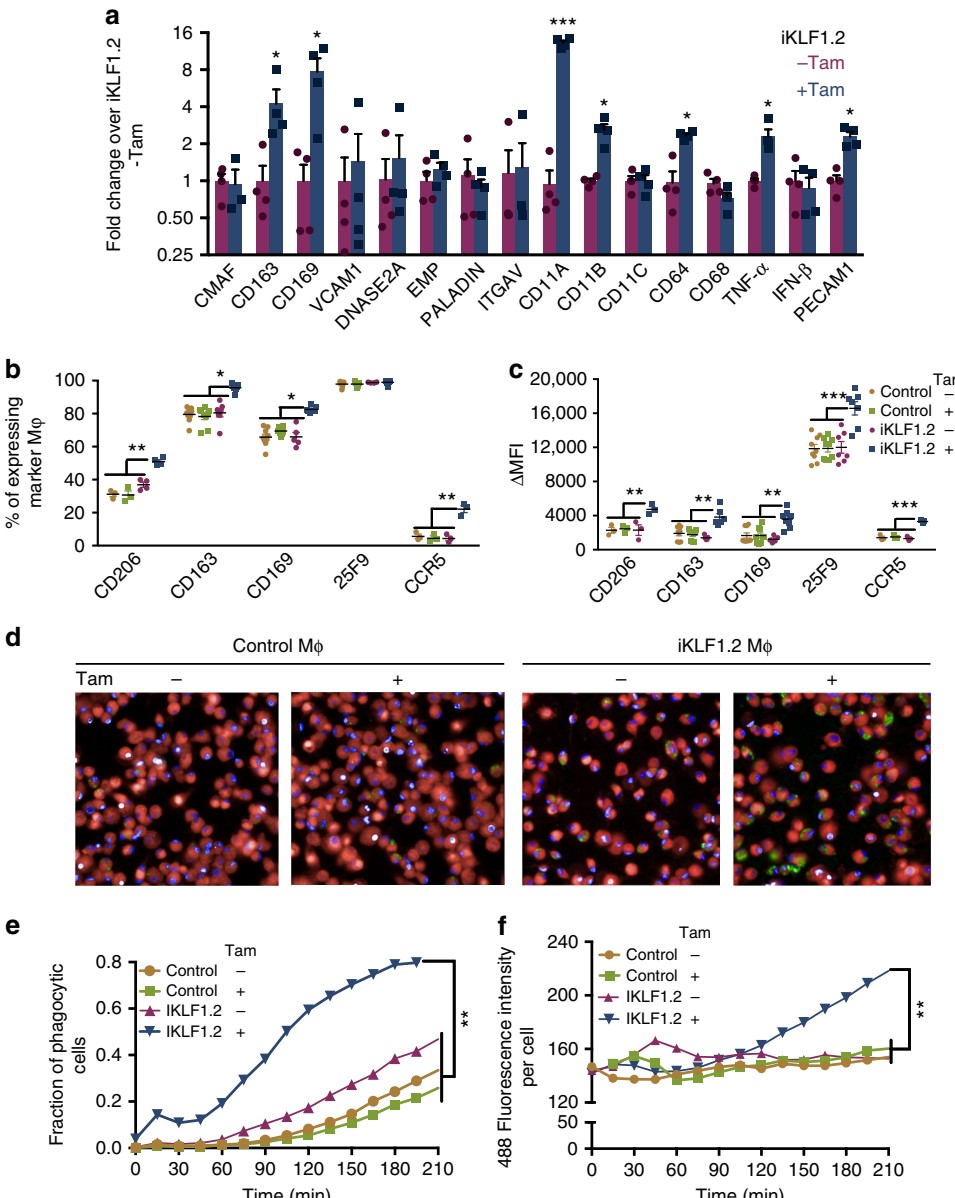

**Fig. 2** Activation of KLF1 in iPSC-DMs alters macrophage phenotype. **a** Real time PCR analyses of EI macrophage related genes in iPSC-DM from iKLF1.2 iPSCs in the presence and absence of tamoxifen (Tam) ($n = 4$, non-parametric Wilcoxon test). **b** Flow cytometry analyses of EI-related cell surface markers expression in control iPSC-DMs (control) and iKLF1.2-DMs (iKLF1.2) in the presence and absence of tamoxifen ($n = 4$ biologically independent samples, non-parametric Kruskal–Wallis test and Dunn's post-test). **c** Mean fluorescence intensity (MFI) of cell surface marker expression in parental iPSC-DMs (control) and iKLF1.2-DMs in presence and absence of tamoxifen ($n = 4$ biologically independent samples, non-parametric Kruskal–Wallis test and Dunn's post-test). **d** Images captured at 175 min after addition of Zymosan-green beads to control iPSC-DMs (Control Mϕ) and iKLF1.2 DMs (iKLF1.2 Mϕ) cells in the presence and absence of tamoxifen (×40). **e** Phagocytic fraction analyses as measured by the proportion of cells containing green beads from 0 to 210 min in control and iKLF1.2-DMs (−/+tamoxifen) ($n = 5$ biologically independent samples, two-way ANOVA and Bonferoni post-test). **f** Phagocytic index as calculated by level of green fluorescence per phagocytic cell ($n = 5$ biologically independent samples, two-way ANOVA and Bonferoni post-test). *$p < 0.05$, **$p < 0.01$, ***$p < 0.001$, ****$p < 0.0001$

monocyte-derived macrophages (MDMs). As *MAF* is also reported to be a marker for yolk sac macrophages, this supports the idea that the phenotype of iPSC-DMs is comparable to tissue resident macrophages[17,18]. *KLF1* was expressed at lower levels in iPSC-DMs compared to MDMs (Fig. 1a), and as *EKLF (KLF1)* had been implicated in the function of murine EI macrophages[22], we hypothesized that enhancing the level of *KLF1* might direct iPSC-DMs into a more EI-like phenotype.

**Generation of macrophages from the iKLF1.2 iPSC line.** We established an iPSC line carrying an inducible *KLF1-ER*[T2] transgene targeted to the safe harbour *AAVS1* locus (herein named iKLF1.2) (Fig. 1b)[20]. IPSC-DMs were generated in a stepwise protocol by first generating embryoid bodies (EBs) in the presence of VEGF, BMP4 and SCF then EBs were transferred to gelatin-coated plates and cultured in presence of IL-3 and CSF1. From day 16, myeloid progenitor cells were harvested from the

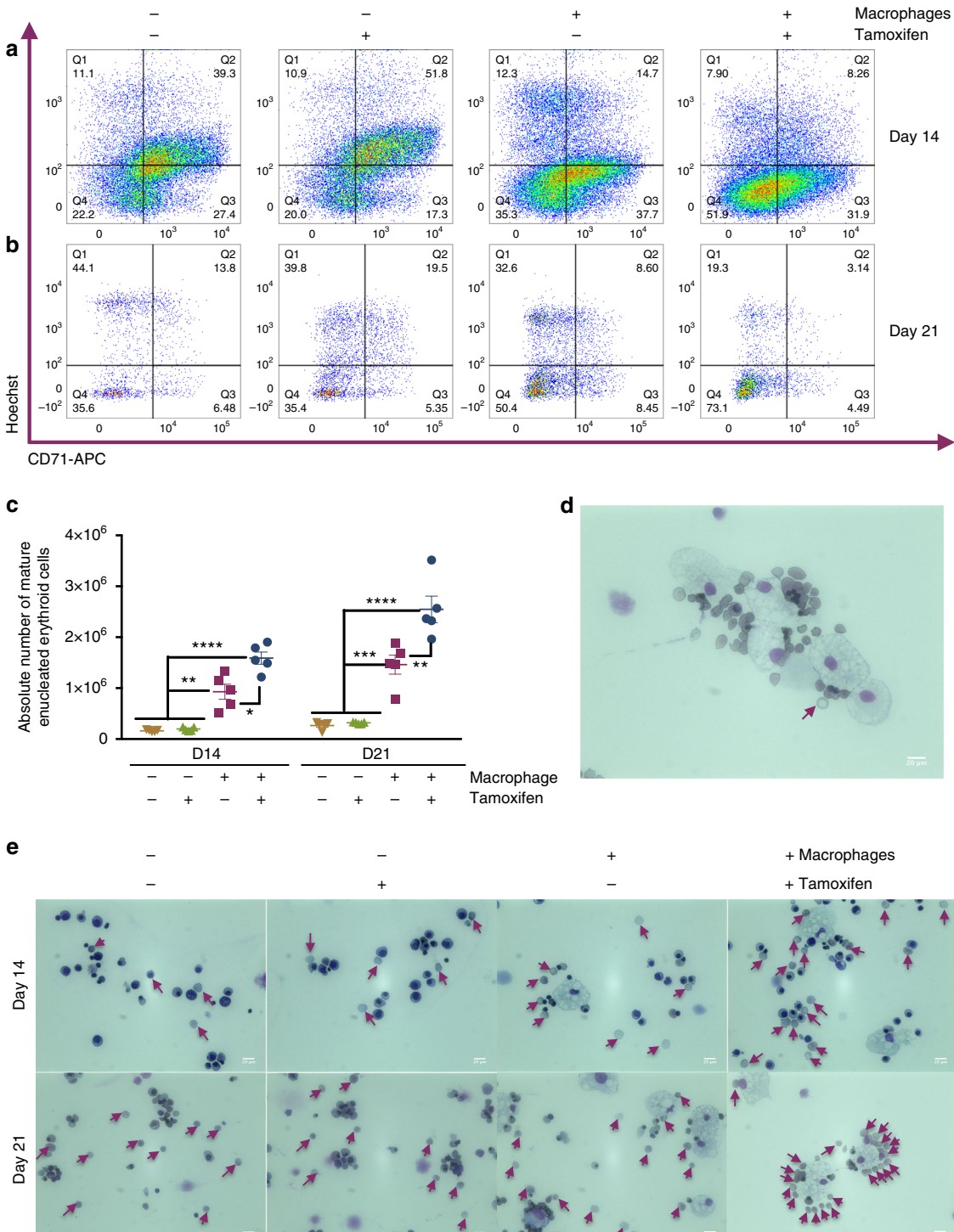

**Fig. 3** KLF1-activated iPSC-DMs enhance erythroid maturation. **a**, **b**. Flow cytometry analyses of live CD235a[+]-gated cells of UCB-CD34[+] erythroid cells cultured alone or in co-culture with iKLF1.2-DMs in the presence and absence of tamoxifen at day 14 (**a**) and day 21 (**b**) stained with anti-CD71 antibody and Hoechst dye (see Supplementary Fig. 3 for gating strategy and FMO controls). **c** Quantification and statistical analysis of absolute numbers of fully mature enucleated erythroid cells of replicate co-culture experiments ($n = 5$ biologically independent samples; two-way ANOVA with Tukey's post-test). **d** Cytospin of co-culture of KLF1-activated macrophages and UCB CD34[+] cells showing close association and a fully mature erythroid cell with biconcave shape (arrow) (scale bar, 20 μm). **e** Cytospins of UCB CD34[+] erythroid cells cultured alone or in co-culture with iKLF1.2-DM in the presence and absence of tamoxifen at day 14 (upper panels) and day 21 (lower panels); arrows point to enucleated cells (scale bar, 20 μm). $*p < 0.05$, $**p < 0.01$, $***p < 0.001$, $****p < 0.0001$

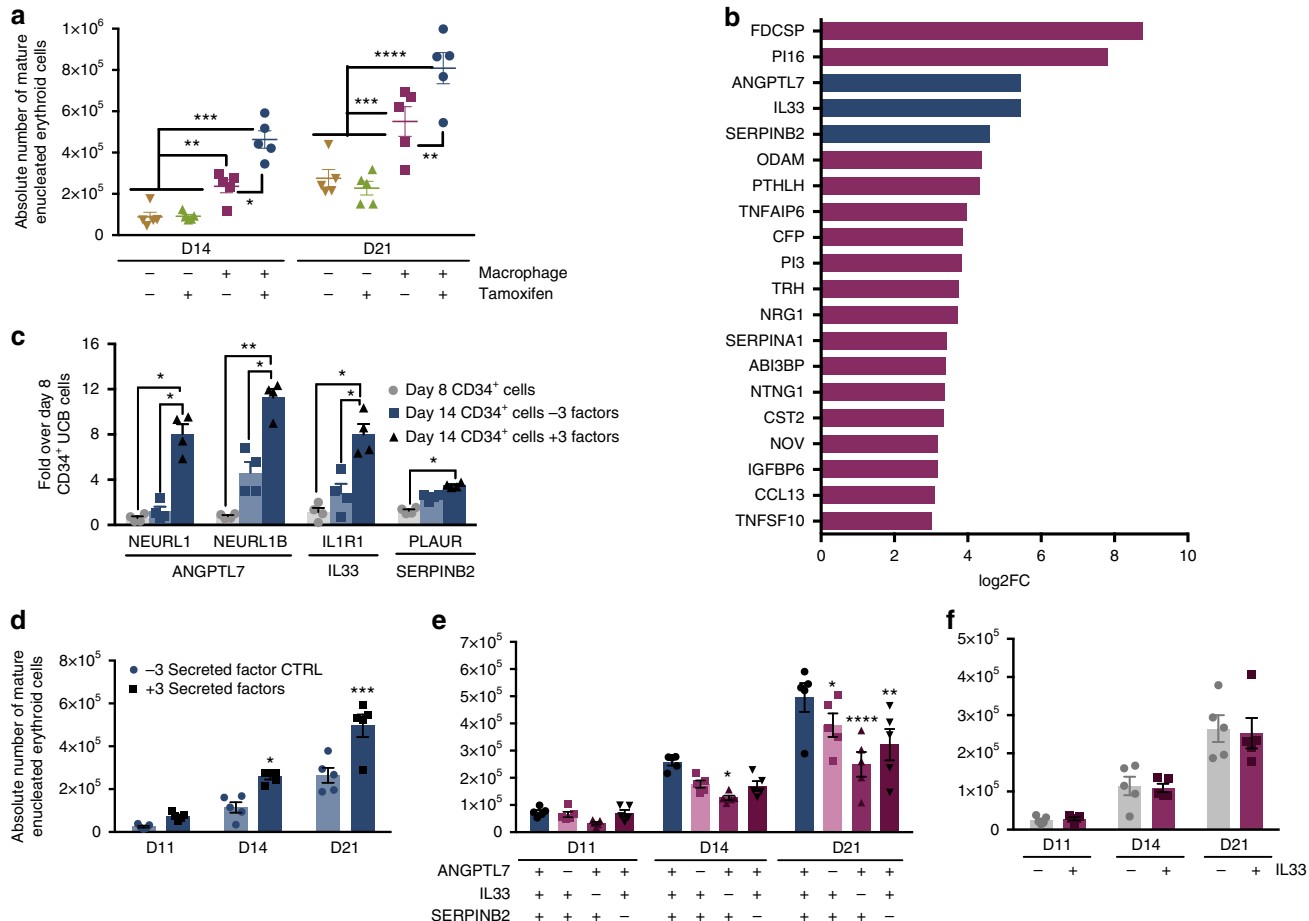

**Fig. 4** Paracrine mechanism involved in erythroid maturation. **a** Quantification and statistical analysis of the absolute number of fully mature enucleated erythroid cells in replicate experiments where cells were differentiated alone or in the presence of iKLF1.2-DMs (−/+ tamoxifen) where cell contact was inhibited using a trans-well ($n = 5$ biologically independent samples; two-way ANOVA with Tukey's post-test). **b** Validated top-20 most differentially expressed genes encoding secreted factors that were upregulated by KLF1 in iKLF1.2-DMs categorized by descending log2 fold change in expression ($p < 0.05$). **c** Quantitative PCR analyses of cytokine (ANGPTL7, IL33 and SERPINB2)—related receptors in UCB CD34+ cells at day 14 of erythroid differentiation with and without the addition of all three cytokines ($n = 4$ biologically independent samples, one-way ANOVA with Sidak's Multiple Comparisons). **d** Quantification and statistical analysis of the absolute number of fully mature enucleated erythroid cells in replicate experiments where UBC-CD34+ cells were differentiated in control conditions (−3 secreted factors (CTRL)) or in the presence of ANGPTL7, IL33 and SERPINB2(+3 secreted factors), at days 11, 14 and 21 ($n = 5$ biologically independent samples, two-way ANOVA with Dunnette's post-test). **e** As above (**d**), under culture conditions where each cytokine was excluded ($n = 5$ biologically independent samples, two-way ANOVA with Dunnette's post-test). **f** Quantification and statistical analysis of the absolute number of mature, enucleated erythroid cells in replicate experiments where UBC-CD34+ were differentiated in the absence and presence of IL33 alone ($n = 4$ biologically independent samples, two-way ANOVA). *$p < 0.05$, **$p < 0.01$, ***$p < 0.001$, ****$p < 0.0001$

supernatant and cultured in CSF1 for 9–12 days[18]. Tamoxifen was added to iPSC-DMs for the last 4 days to induce nuclear translocation and therefore activation of the KLF1 protein. Macrophages derived from the iKLF1.2 iPSC line had a comparable morphology to those derived from the control, parental iPSC line (known as SFCi55) (Fig. 1c, Supplementary Fig. 1A) and this was not affected by the addition of tamoxifen (Fig. 1c). As expected, macrophages generated from the iKLF1.2 iPSC line demonstrated a higher expression of *KLF1* mRNA expression compared to macrophages derived from monocytes or control iPSC-DMs (Fig. 1d). Using an anti-HA antibody, we demonstrated that the KLF1-ER[T2] fusion protein is expressed in the cytoplasm of iKLF1.2-DMs and translocates to the nucleus upon tamoxifen addition thus demonstrating that the fusion protein activation strategy can function in terminally differentiated iPSC-DMs and that the *KLF1-ER[T2]* transgene is not silenced (Supplementary Fig. 1B).

**Activation of KLF1 in iPSC-DMs alters macrophage phenotype.** We assessed the effect of KLF1 activation on the mRNA expression of previously reported *KLF1* target genes including *VCAM1*, *DNASE2A*; cell adhesion molecules involved in macrophage–erythroblast interaction (*ITGAV*, *EMP/MAEA*, *PECAM1*, *CD163* and *CD169*); EI macrophage markers (*CD64*, *CD68*, *CD11A*, *CD11B*, *CD11C*) and extrinsic regulators of erythropoiesis (*PALD*, *IFN-* and *TNF-α*)[16,22–28]. Activation of KLF1 by the addition of tamoxifen resulted in the increased expression of a subset of these transcripts including *CD163*, *CD169*, *CD11A*, *CD11B*, *CD64*, *TNFα* and *PECAM1* in macrophages derived from iKLF1.2 iPSCs (Fig. 2a). These genes were not upregulated upon tamoxifen addition to control iPSC-DMs confirming that this change in phenotype was associated with KLF1 activation rather than a non-specific effect of tamoxifen (Supplementary Fig. 1C). KLF1 activation increased the proportion of iPSC-DMs expressing EI-associated cell surface markers (CD206, CD163, CD169

and CCR5) (Fig. 2b) and the expression level of these markers as measured by the mean fluorescent intensity (MFI) (Fig. 2c). IPSC-DMs expressed the marker 25F9 irrespective of tamoxifen treatment but the level of expression per cell was significantly higher after KLF1 activation (Fig. 2b, c). To assess whether activation of KLF1 altered iPSC-DM function, we used a quantitative assay of phagocytic activity[29]. Using a live cell imaging strategy[29], we measured the proportion of cells that had phagocytosed and the phagocytic activity of each cell with the intensity of green fluorescence per cell correlating with the number of ingested beads (Fig. 2d). Both the proportion of phagocytic cells and the phagocytic activity per cell of tamoxifen-treated, iKLF1.2-DMs was significantly higher compared to all controls including control iPSC-DMs and iKLF1.2-DMs in the absence of tamoxifen (Fig. 2e, f, respectively). To demonstrate the reproducibility of this strategy to alter the functional phenotype of macrophages by genetic programming, we generated macrophages from three independently derived iPSC lines (iKLF1.6, iKLF1.7 and iKLF1.12). Activation of KLF1 by the addition of tamoxifen to all three cell lines resulted in a comparable increase in phagocytic activity and an increase in expression of EI-related genes (Supplementary Fig. 2A–C).

**KLF1-activated iPSC-DMs enhance erythroid maturation**. Umbilical cord blood (UCB)-derived CD34+ HPCs can produce enucleated RBCs in vitro with varying efficiency[30]. We hypothesized that their maturation and enucleation efficiency could be enhanced by co-culture with 'EI-like' iPSC-DMs. CD34+ UBC-derived HPCs were cultured in SCF, EPO, IL3 and hydrocortisone for 7 days, then from day 8, the cells were co-cultured with iKLF1.2 iPSC-DM in SCF, EPO and transferrin with or without tamoxifen. Mature, enucleated cells were identified at day 14 and day 21 by assessing the number of CD235a-expressing erythroid cells that were negative for CD71 and the nuclear, DNA dye, Hoechst [18] (Supplementary Fig. 3A). The enucleated phenotype of CD71−Hoechst− cells was confirmed by cell sorting and comparing their morphology to Hoechst+ cells in cytospin preparations (Supplementary Fig. 3B). The absolute number of mature enucleated erythroid cells (CD235a+, CD71−, Hoechst−) was higher in cells that were co-cultured with macrophages and this was further increased when co-cultured with iKLF1.2-DMs- that had been treated with tamoxifen (Fig. 3a, b). Quantification of at least five replicate experiments demonstrated that co-culture with iPSC-DMs resulted in a fourfold increase in the absolute number of mature, enucleated cells. This was further increased to 10-fold when co-cultured with KLF1 -activated macrophages (Fig. 3c, Supplementary Fig. 4E). The increase in absolute number of cells is due to an increase in overall cell proliferation and viability between days 8 and 21 as well as a significant increase in the terminal differentiation of CD235a+ erythroid cells (Supplementary Fig. 4). Quantitative RT-PCR analyses indicated that macrophage co-culture did not alter erythroid specification of CD34+ cells because the relative level of expression of GPYA, KLF1, BCL11A and GATA1 were unaffected as was the globin profile (Supplementary Fig. 5A, B). In contrast, co-culture with KLF1-activated macrophages had a significant effect on the level of expression of the terminally differentiated erythroid markers, DMTN, AHSP and ANK1 supporting our flow cytometry data (Supplementary Fig. 5C). Morphological analyses confirmed the increase in the proportion of enucleated cells in a qualitative manner and also demonstrated a close association between macrophages and differentiating erythroid cells (Fig. 3d, e).

**Paracrine mechanism involved in erythroid maturation**. To assess whether cell contact was required for the enhanced

maturation in iPSC-DM co-cultures, we used a trans-well assay (0.4 μm pore size diameter) where media and secreted factors could be exchanged but direct macrophage-erythroid cell contact was prevented. The trans-well culture setup reduced the average baseline level of enucleation but we noted a significant increase in the absolute number of enucleated cells when KLF1-activated macrophages were present on the other side of the trans-well (Fig. 4a, Supplementary Fig. 6A, B). KLF1-activated macrophages increased the absolute number of enucleated cells by five- and threefold at day 14 and 21, respectively, and this was significantly higher than control macrophages (Supplementary Fig. 6H). We noted that the overall fold increase is not as high as that observed in contact culture (Fig. 3) suggesting that both cell–cell contact and secreted factors are involved in mediating the effects of KLF1. Interestingly, when contact was inhibited we observed no effect of KLF1 activation on the proliferation and viability of erythroid cells compared to iPSC-DMs in the absence of tamoxifen (Supplementary Fig. 6D, E) and thus the phenotypic effect of KLF1 activation appeared to be restricted to an effect on terminal differentiation (Supplementary Fig. 6G).

**KLF1 upregulates cell communication and protein-binding factors**. To identify KLF1 target genes in iKLF1.2-DMs that are potential mediators of the observed effect on erythroid maturation, we carried out RNA sequencing of iKLF1.2-DMs in the presence and absence of tamoxifen. We identified 803 and 593 genes that were up- and downregulated upon KLFI activation, respectively (Supplementary Fig. 7A, B). Gene ontology analysis was performed on 803 upregulated genes using the Panther ontology web-tool[31–33]. When genes were classified according to molecular function, 35% of them were associated to a binding function and of those, over 75% fell into a protein-binding category with the vast majority being annotated as receptor binding. When classified according to biological function, the largest fraction (25%) fell into the category of cellular process with the majority being annotated as cell–cell communication-related genes (Supplementary Fig. 7C, D).

We validated the top-15 upregulated genes by qRT-PCR of RNA isolated from control and iKLF1.2-DMs (−/+tamoxifen) and demonstrated that the upregulation was due to KLF1 activation and not the addition of tamoxifen (Supplementary Fig. 7E, F).

**Identification and characterization of secreted factors**. As co-culture of KLF1-activated macrophages had a significant effect on the maturation and enucleation of co-cultured UCB-CD34+ erythroid cells when contact was prevented, we focused on characterizing KLF1-upregulated (>3 log2 fold change) genes that encode secreted factors. We first validated their upregulation by qRT-PCR and identified 20 genes encoding secreted factors that were upregulated by KLF1 activation in iKLF1-DMs but not upon the addition of tamoxifen in control iPSC-DMs (Supplementary Fig. 8A, B).

For functional studies we selected genes encoding the secreted factors ANGPTL7, IL33 and SERPINB2 that, at the time of study, were the top-3 secreted factors commercially available as human recombinant proteins that had been functionally validated (Fig. 4b). By qRT-PCR we validated that they were also upregulated in macrophages derived from the iKLF1.6, iKLF1.7 and iKLF1.12 iPSC lines (Supplementary Fig. 8C). We first assessed whether differentiating UCB-CD34+ cells had the potential to be responsive to these selected cytokines by testing whether their putative receptors were expressed (Fig. 4c). Genes encoding the receptors for ANGPTL7 (NEURL1 and NEURL1), IL33 (IL1R1) and SERPINB2 (PLAUR) were expressed in

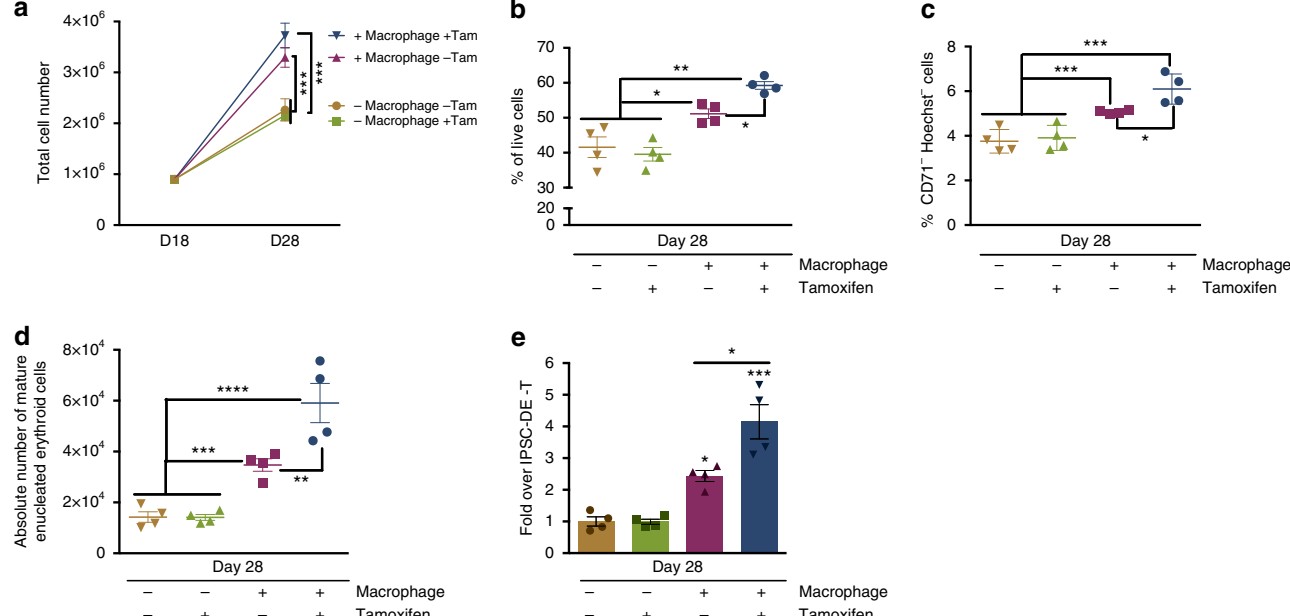

**Fig. 5** KLF1 activation enhances iPSC-derived erythroid cell production. **a** Total cell number of iPSC-derived erythroid cells cultured alone or in co-culture with iKLF1.2-DMs in the presence and absence of tamoxifen (Tam) from day 18–28 of differentiation ($n = 4$ biologically independent samples, one-way ANOVA with Tukey's post-test). **b** Viability of iPSC-derived erythroid cells at day 28 following above culture conditions ($n = 4$ biologically independent samples, one-way ANOVA with Tukey's post-test). **c** Live CD235a+-gated iPSC-derived erythroid cells that are negative for CD71 and Hoechst staining following above culture conditions ($n = 4$ biologically independent samples; one-way ANOVA with Tukey's post-test). **d** Quantification and statistical analysis of the absolute number of mature, enucleated erythroid cells in replicate experiments as described above ($n = 4$ biologically independent samples; one-way ANOVA with Tukey's post-test). **e** Fold change of mature, enucleated iPSC-derived erythroid cells following co-culture with iKLF1.2-DMs (−/ +tamoxifen) compared to iPSC-derived erythroid cells cultured alone (− tamoxifen) ($n = 4$ biologically independent samples; one-way ANOVA with Tukey's post-test). *$p < 0.05$, **$p < 0.01$, ***$p < 0.001$, ****$p < 0.0001$

differentiating (day 14) UCB-CD34+ cells and the level was higher when cells were cultured in the presence of the three cytokines (Fig. 4c).

We next assessed the combined effect of ANGPTL7, IL33 and SERPINB2 on the differentiation and maturation of UCB-CD34+ cells in the absence of macrophages. When all three cytokines were included in the differentiation protocol, the absolute number of mature, enucleated cells (CD235a+ CD71−Hoescht−), was significantly higher compared to control cultures and this was apparent at all stages of the differentiation protocol (day 11, 14 and 21) (Fig. 4d). To assess the contribution of individual cytokines, we used an elimination strategy and calculated the absolute number of mature cells at day 11, 14 and 21 (Fig. 4e). At day 14, the number of cells was reduced when IL33 was removed compared to cultures with all three cytokines (Fig. 4e). We noted that the decrease was specifically due to the loss of CD71 expression at a stage prior to the enucleation process (Supplementary Fig. 9A, B) suggesting that IL33 enhanced the timing of maturation. At day 21, the highest number of CD71− enucleated cells was observed in cultures when all three cytokines were included and a significant reduction was observed when each one of the three cytokines was removed. Statistical analyses indicate that IL33 is the most important player because the effect of its removal at day 21 was more significant ($p < 0.0001$) than the removal of either ANGPTL7 ($p = 0.0259$) or SERPINB2 ($p = 0.0089$) (Fig. 4e). Based on these findings we tested the effect of IL33 alone, but surprisingly the absolute number of enucleated cells was comparable to control cultures with no secreted factors added, suggesting that the factors act in synergy (Fig. 4f).

**Functional assessment of cultured RBCs**. To confirm that our novel culture methods did not affect the functionality of the resultant erythroid cells, we assessed their ability to bind and release oxygen. UCB-CD34+ cells that were differentiated for 21 days in co-culture with macrophages in the presence and absence of tamoxifen or in the presence of ANGPTL7, IL33 and SERPINB2 had comparable $O_2$ association and dissociation curves compared to control cultures and to adult peripheral blood erythrocytes (Supplementary Fig. 10).

**KLF1 activation enhances iPSC-derived erythroid cell production**. We next assessed whether EI-like iPSC-derived macrophages could enhance the production of erythroid cells from the limitless iPSC source. We previously demonstrated that activation of the transcription factor KLF1 enhanced the maturation of iPSC-derived erythroid cells but we had not considered that part of the effect could have been due to the effect of KLF1 within the niche of the heterogenous population of differentiating iPSCs[20]. We used a four-step erythroid differentiation protocol involving mesoderm specification (day 0–3), haematopoietic induction (day 4–14), erythropoietic induction (day 15–21) and erythroid maturation (day 22–28) as described[8,20]. We demonstrate that co-culture with control iPSC-derived macrophages from day 18 of erythroid differentiation, increased the absolute number of mature enucleated erythroid cells by twofold. When co-cultured with KLF1-activated macrophages this increased to fourfold (Fig. 5). The increase in the absolute number of mature erythroid cells is, in part, due to an increase in cell proliferation and viability (Fig. 5a, b) as well as an effect on maturation per se (Fig. 5c–e).

## Discussion

The production of RBCs in vitro could solve problems associated with blood transfusion such as limitations in supply, transfusion transmitted infections, and immune incompatibility. Culture protocols have been developed for the production of RBC in vitro from human CD34[+] HPCs, PSCs and immortalized erythroid progenitor cell lines[7–12]. However, regardless of the cell source, the protocols are relatively inefficient and a variable number of cells from the different sources undergo the enucleation process that marks the final steps of erythroid maturation. Development of improved protocols to generate mature RBCs in vitro will benefit from a deeper understanding of the cellular and molecular interactions involved erythroid maturation, particularly in the communication between erythroid island macrophages and the surrounding developing erythroblasts.

Attempts to recreate the human EI niche in vitro have involved co-culture with monocyte-derived macrophages (MDMs) with the addition of glucocorticoid promoting their differentiation into an EI-like phenotype and promoting erythroblast expansion and survival[14,15]. Our strategy represents a significant improvement in the modelling of the EI niche in vitro for a number of reasons. Firstly, IPSC-derived macrophages have been reported to have a tissue resident-like phenotype[17,18] and in keeping with this, we noted that iPSC-DMs have a higher level of *MAF* compared to MDMs. Secondly, iPSC-DMs provide a limitless source of either healthy or patient-derived macrophages and thus are not dependent on repeated collection of peripheral blood monocytes. Finally, the ability to expand iPSCs indefinitely makes then amenable to the vast arsenal of genetic tools, including over-expression and gene targeting strategies that allows for the testing of novel factors involved in their action.

We noted that KLF1 was expressed at a low level by both iPSC-DMs and MDMs and, given that KLF1 has been implicated in the murine EI niche, we speculated that enhanced expression of this transcription factor might programme macrophages into an EI-like phenotype. We used an inducible KLF1-ER[T2] transgene targeted to the *AAVS1* locus to allow temporally controlled activation during macrophage differentiation. We have previously shown that the production and function of macrophages from iPSCs is unaffected by *AAVS1* targeting per se and is resistant to epigenetic silencing[29]. Thus, this provides an ideal platform for testing the specific effect of transcription factor activation that can be applied to any iPSC line of choice. Activation of KLF1 in four independently derived iPSC lines carrying an inducible KLF1-ER[T2] increased the expression of EI-associated genes and cell surface markers. Interestingly, not all the genes previously identified as KLF1 targets are activated in this system, which likely reflects the fact that transcriptional control by KLF1 is context dependent and its numerous known protein partners are cell type specific. We demonstrate that KLF1-activated macrophages have an enhanced rate of phagocytosis, a feature of EI macrophages that has presumably evolved in the first instance to clear free nuclei. In our co-culture experiments, KLF1-activated EI-like macrophages increased the production of mature and enucleated erythroid cells from human UCB-CD34[+] HPCs and iPSCs, implying that EI macrophages have also evolved to provide a supportive and/or instructive role for maturing RBCs. Indeed, this is consistent with studies in the murine system where an extrinsic role for KLF1 in the EI niche has been reported[21,22].

We demonstrate that erythroid maturation of UCB-CD34[+] HPCs can be enhanced to some extent when cell contact was inhibited indicating that this action of KLF1 target genes is in part mediated by secreted factors. We identified a number of secreted factors that were upregulated in KLF1-activated macrophages and have shown that a combination of three of these, ANGPTL7, IL33 and SERPINB2 had a significant impact on the maturation of UBC-CD34[+] HPCs. The presence of all three cytokines provided the best maturation conditions with removal of each of the cytokines resulting in a significant reduction in the number of mature cells. IL33 appeared to be the most important player because its removal resulted in the most significant decrease but addition of IL33 alone did not enhance the maturation process. This indicates that IL33 must act in concert with the other cytokines and/or amplify their signals in keeping with studies that have reported an amplifying effect of IL33 on macrophage polarization[34]. Interestingly, IL33 is expressed in erythroid progenitor cells and released during haemolysis [30][35]. In light of our result, we propose that IL33 could be amplifying the action of other cytokine signalling pathways providing a feedback mechanism to stimulate the maturation of new RBCs. We have shown that the IL33 receptor (IL1R1) is expressed in differentiating UCB-CD34[+] cells and upregulated upon addition of IL33. Furthermore, IL1R1 and a predicted downstream target, GATA3[36], were among the most upregulated genes when KLF1 was activated (Supplementary Fig. 11). Together, these findings suggest that IL-33 plays a role in the erythroblastic island niche, in both erythroid cells and in the central macrophage possibly activating different downstream targets in the two cell types.

Many angiopoietin-like proteins are expressed by stromal cells and involved in supporting the activity and engraftment of HSPCs with ANGPTL7 being shown to increases the expansion of human HSPCs ex vivo[37]. In the murine system, *Angptl7*-deficient HSPCs were able to repopulate irradiated recipients but *Angptl7*-null mice were unable to provide the supportive environment for wild-type HSPC transplantation demonstrating an extrinsic rather than a cell-autonomous role[38]. A role for ANGPTL7 in HSPC differentiation has not been reported, but our experiments indicate its role in the maturation and differentiation of the erythroid lineage.

The serine protease inhibitor, SERPINB2, is a coagulation factor known to be present in macrophages and plays a key role in preventing apoptosis[39]. In the murine system *Serpinb2*$^{-/-}$ macrophages promote IFN-ϒ production and secretion[40]. IFN-β activation contributes to the impaired erythropoiesis of *Klf1* and *DNAse1* deficient mice[41,42] thus predicting that the addition of SERPINB2 would reduce IFN-β expression. However, we were unable to test this hypothesis because IFN-β transcripts were not detectable in UCB-CD34[+] erythroid cells.

In summary, we report that human macrophages can been manipulated through the enforced expression of a transcription factor resulting in significant alterations in their phenotype. The *AAVS1* targeting strategy provides a solid reproducible platform for the introduction of genes or factors that are predicted to modulate or stabilize macrophage phenotype and function. We have used this system to establish an in vitro model of the human erythroblastic island niche and to study the molecular processes involved in the final steps of erythroid maturation that are otherwise inaccessible to study. Future application of this technology could allow the generation of tissue-resident macrophages associated with other tissues using transcription factors that have been defined for these populations[43].

## Methods

**Maintenance of human iPSC lines**. The human iPSCs lines SFCi55 (parental, control) and SFCi55-iKLF1.2 were generated in house[20]. The SFCi55 iPSC line was originally generated using fibroblasts obtained from blood group O Rhesus negative individuals by R Biomedical under REC 1/AL/0020 ethical approval and programmed to iPSCs using Yamanaka factors on episomal vectors. Both lines were confirmed to be pluripotent and have normal karyotype[20]. All IPSC lines were routinely tested for mycoplasma and maintained in StemPro hESC SFM media (Gibco) with 20 ng/ml bFGF (R&D) (Maintenance media). Wells were pre-coated with CELLstart (Gibco) for 1 h before. Cells were passaged using the StemPro EZPassage tool (Thermo Fisher Scientific). Media change was performed every day

and cells passaged at a ratio of 1:4 every 3–4 days, when cells reached 70% confluency.

**Generation of human iPSC-derived macrophages.** We adapted the previously published differentiation protocol that resulted in optimal macrophage production[18,44]. Maintenance media on one confluent well of a six-well plate was replaced with 1.5 ml of day 0 mix, which consisted of StemPro hESC SFM (Gibco) supplemented with BMP4 (50 ng/ml), VEGF (50 ng/ml) and SCF (20 ng/ml). Colonies were cut using the EZPassage tool into two wells with 2.25 ml of day 0 mix cultured in Ultralow Attachment 6-well plates (Greiner) for 4 days to induce embryoid body (EB) formation. In total, 10–15 EBs were transferred to each gelatin-coated well tissue-culture-grade six-well plates in 3 ml day 4 mix (X-VIVO15 media supplemented with M-CSF (100 ng/ml), IL3 (25 ng/ml), Penicillin-Streptomycin, Glutamax (2 mM) and β-mercaptoethanol (0.055 mM) and media was changed every 3–4 days. After 3 weeks, non-adherent monocyte-like precursors were harvested from the supernatant and plated into untreated bacteriological plates or six-well plates in Maturation mix (X-VIVO15 media supplemented with M-CSF (100 ng/ml), Glutamax (2 mM) and Penicillin-Streptomycin (1%)) for 9–11 days. Harvesting was repeated very 3–4 days for up to 3 months. To activate KLF1 in iKLF1.2-DMs, tamoxifen (100 nM) was added to the adherent macrophage population for the last 4 days of the differentiation process.

**Culture of umbilical cord blood-derived CD34+ cells.** Frozen Umbilical cord blood (UCB)-derived CD34+ cells were purchased from Stemcell Technologies (Cat No. 70008.5) from consenting donors with protocols approval by either the Food and Drug Administration (FDA) or an Institutional Review Board (IRB). UCB cells were expanded and differentiated using a stepwise protocol[12,45]. Briefly cells were cultured ($1–6 × 10^4$ cells/ml) in ISHI base media (Iscove's basal media (Biochrom AG), human $AB^+$ serum (5%), heparin (3 U/ml) and insulin (10 μg/ml)) supplemented with SCF (60 ng/ml), IL3 (5 ng/ml), EPO (3 U/ml), Hydrocortisone (1 μM) and holo-Transferrin (200 μg/ml) for 6 days when this expanded cell population could be frozen in batches of $10^6$ cells/ml in a mix of ISHIT media (60%), knockout serum replacement (30%) and DMSO (10%). Upon thawing, cells were recovered in the above media (day 0–6) for 2 days (now considered 'day 8'). Cell density was adjusted to $10^5$ cells/ml in ISHI media supplemented with SCF (10 ng/ml), EPO (3 U/ml), and holo-Transferrin (300 μg/ml), cultured for a further 3 days then cultured at a density of $10^6$ cells/ml in ISHI medium supplemented with EPO (3 U/ml) and holo-Transferrin (300 μg/ml) until day 21. Media was changed every 3–4 days throughout the protocol. The same media was used in co-culture experiments that were set up from day 8 at a ratio of 3:1 (iPSC-DM:CD34+-derived cells). To test cell contact vs. paracrine mechanisms, we used trans-wells with 0.4 μm diameter pore size (Bioscience). For experiments where secreted factors were tested, they were added from day 8 onwards every 2 days. The final concentration of ANGPTL7 (Preprotech Cat# 130-22), IL33 (Preprotech Cat# 200-33) and SERPINB2 also known as PAI 2 (Preprotech Cat# 140-06) were 60, 75 and 75 ng/ml, respectively.

**Erythroid differentiation of iPSCs.** iPSCs were differentiated in a stepwise manner consisting of four stages: mesoderm specification, haematopoiesis induction, erythropoiesis induction and erythroid maturation as described previously[20]. Maintenance media on one confluent well (six-well plate) was replaced with 1.5 ml of Stemline II medium (Sigma), supplemented with 10 ng/ml BMP4 (R&D), 10 ng/ml VEGF (R&D), 10 ng/ml Wnt3A (R&D) and 5 ng/ml Activin A (R&D), and 2 μM GSK-3β inhibitor VIII (Merck). Colonies were cut using the EZPassageTM, then transferred to two wells of a low attachment plate (Greiner) to induce EB formation. EBs were topped up with 0.5 ml of Stemline II media with same cytokine concentrations. EBs were then dissociated to a single cell suspension using StemPro Accutase (Thermo Fisher) on day 3. Cells were seeded at a cell density of $2 × 10^5$ cells per well in 3 ml of Stemline II hematopoietic stem cell expansion media supplemented with 20 ng/ml BMP4, 30 ng/ml VEGF, 10 ng/ml FGFa, 30 ng/ml SCF, 10 ng/ml IGF2 (R&D), 10 ng/ml TPO (R&D), 5 μg/ml Heparin (Sigma), 50 μM IBMX (Sigma) and 0.4 ng/ml β-estradiol. Cytokines were topped up at days 5, 7 and 9. Differentiating suspension cells were harvested at day 10 and seeded at a density of $1 × 10^5$ cells/ml in Stemline media in the presence of 50 ng/ml SCF, 16.7 ng/ml Flt3L (Preprotech), 6.7 ng/ml IL-3, 6.7 ng/ml (Preprotech), 3 U/ml EPO and 50 μM IBMX. Cytokines were topped up at days 12, 14 and 16. At day 18, differentiating cells were cultured at a density of $5 × 10^5$ cells/ml in IBIT media, made up by 240 ml of Iscove's basal medium (Merck Millipore) supplemented by 1% BSA (Gibco), 10 μg/ml insulin (Sigma), 0.2 μg/ml transferrin and 0.5 ml of β-mercaptoethanol; supplemented with 20 ng/ml SCF, 20 ng/ml IGF1 (Preprotech), 6.7 ng/ml IL3, 6.7 ng/ml IL11 and 3U/ml EPO. Fresh cytokines were topped up in 0.5 ml of IBIT medium per well of a six-well plate at days 20, 22 and 24. At day 25, the differentiating cells were harvested and reseeded at a density of $1 × 10^6$ cells/ml in IBIT medium supplemented with 3 U/ml EPO, 6.7 ng/ml IL-1β (R&D), 6.7 ng/ml IL-6 (R&D), 5% AB plasma (Sigma), and 2 ng/ml sodium selenite (Sigma). Cytokines were topped up at day 27. Differentiating cells were co-cultured with iKLF1.2-DMs (−/+tamoxifen) (100 nM) from day 18 at a ratio of 3:1 (macrophage:erythroid cells). Tamoxifen was topped up very 2 days and erythroid maturation status was assessed by flow cytometry at day 28.

**Cytospin preparation and Rapid-Chrome Kwik-Diff staining.** Cytospin preparations of macrophages or erythroid cells were performed by re-suspending $5 × 10^4$ or $1 × 10^5$ cells in 200 μl of PBS, respectively. Cells were cyto-centrifuged onto polylysine slides at $72 × g$ for 8 min in a Thermo Shandon Cytospin 4, air-dried overnight then stained according to manufacturer's instructions (Thermo Fisher #9990702).

**Immunocytochemistry.** To assess HA-KLF1-ER^T2 sub-cellular localization, $6 × 10^4$ macrophages were cultured in a gelatin-coated Nunc® Lab-Tek® Chamber Slide System (Sigma), fixed in 4% PFA in PBS at room temperature for 10 min, permeabilized in PBS-T (PBS with 0.4% Triton-X100) for 20 min and blocked in PBS-T with 1% BSA and 3% goat serum for 2 h and incubated in anti-HA 1:500 (Clontech #631207) for 1.5 h. Cells were washed with PBS-T thrice for 15 min, incubated in Alexa488 anti-rabbit 1:1000 (Thermo Fisher Scientific #A-11008) for 1.5 h in the dark, washed with PBS-T thrice for 15 min then counter-stained with DAPI 1:1000 (Sigma) for 5 min.

**Flow cytometry.** Single cell suspensions were prepared using StemPro Accutase Cell Dissociation Reagent (Gibco) and re-suspended in PBS with 1%BSA and 5 mM EDTA. Cells were blocked with MACS FcR Blocking Reagent (#130-059-901) for 40 min on ice according to manufacturer instructions. In total, $1 × 10^5$ cells were washed and stained with appropriate antibodies (Supplementary Table 1) for 20 min at room temperature. Dead cells were gated out using DAPI. To assess enucleation, single cell suspensions were stained with Hoechst33342 1:20 (Thermo Fisher #R37605) for 20 min, washed with PBS 1%BSA and 5 mM EDTA then stained with CD71-APC 1:200 (Thermo Fisher, 17-0719-42), CD235a-FITC 1:1000 (EBioscience #11-9987) and LIVE/DEAD™ Fixable Near-IR Dead Cell Stain 1:100 (Thermo Fisher #L10119) for 20 min at room temperature. Cells were washed with PBS with 1%BSA and 5 mM EDTA and kept on ice prior to analysis using LSR Fortessa Analyser (BD) and FlowJo Software.

**Phagocytosis assay.** In total, $8 × 10^4$ iPSC-DMs were plated in tissue-culture-grade 96-well plates (CellCarrier, Perkin Elmer) at least 2 days before assessing their phagocytic activity as previously described[18]. Briefly cells were stained with the nuclear stain, Hoechst33342 and CellMask™ deep red plasma membrane stain then pHrodo™ Green Zymosan A BioParticles were applied immediately prior to live imaging using the Operetta High-Content Imaging System (Perkin Elmer). The number of iPSC-DMs that had ingested beads (phagocytic cell fraction) and the average number of beads that each cell had ingested (phagocytic index) was quantified using Columbus Image data storage and analysis system.

**Gene expression analyses.** Total RNA extraction was carried out using the RNAeasy Mini Kit (Qiagen); cDNA was generated from 500 ng of total RNA using the high capacity cDNA synthesis kit (Applied Biosystem). Two nanograms of cDNA were amplified per reaction and each reaction was performed in technical triplicates using the LightCycler 384 (Roche) with SYBR Green Master Mix II (Roche). GADPH, β-actin and B2M were used as reference genes and the geometrical mean was used to normalize the data. Primer sequences and efficiencies are reported (Supplementary Table 2). RNA sequencing was performed by Edinburgh Genomics and data deposited in in NCBI's Gene Expression Omnibus (accession number GSE125150)[46,47].

**Hemox analyses.** The oxygen carrying potential of cultured erythroid cells and adult blood was determined using a Hemox analyser (TCS, New Hope, USA) according to manufacturer' instructions. In short, ~$2.5 × 10^7$ pelleted cells or 50 μl of peripheral blood control erythrocytes were diluted in 5 ml HEMOX solution (PBS supplemented with 0.5% human serum albumin and 0.01% Y-30 (A5768, Sigma)). Simultaneous duel wavelength spectrophotometry at 560 and 576 nm was measured to calculate the oxyhemoglobin fraction at different oxygen tensions[48].

**Statistical analyses.** Data are expressed as mean ± standard error mean (SEM). Statistical tests that were used are indicated in figure legends and were performed using Graph Pad software version 6.0c. P-values <0.05 were considered statistically significant (*$p < 0.05$, **$p < 0.01$, ***$p < 0.001$, ****$p < 0.0001$).

**Reporting summary.** Further information on experimental design is available in the Nature Research Reporting Summary linked to this article.

## Data availability

RNA sequencing data that support the findings of this study deposited in NCBI's Gene Expression Omnibus and are accessible through GEO Series accession number GSE125150.

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

## Acknowledgements

We thank Fiona Rossi and Claire Cryer for assistance with flow cytometry, Eoghan O'Duibhir and Bertrand Vernay with microscopy, Edinburgh Genomics and Jon Manning for RNA sequencing and analyses and James Bieker for critical comments on the manuscript. This work was funded by Wellcome Trust (102610) and Innovate UK (L.M.F., A.F. and A.H.T.), CONACYT (M.L.-Y.) and the Scottish Funding Council (C.-T.Y.). L.C. and J.W.P. were supported by Wellcome Trust (101067/Z/13/Z), Medical Research Council (MR/N022556/1), and COST Action BM1404 Mye-EUNITER (http://www.mye-euniter.eu) and E.S., E.v.d.A. and M.v.L were funded by ZONMW 40-41400-98-1327.

## Author contributions

M.L.-Y. designed and performed experiments, analysed data and wrote the paper, A.F. and C.-T.Y. designed and performed experiments and analysed data; A.H.T. helped with cell culture experiments; L.C. provided support and advice on macrophage phenotyping, E.S. performed oxygen dissociation assays, A.M. and J.C.M. provided UCB samples, L.M.F. designed the experiments, analysed data and wrote the manuscript. J.W.P., M.v.L. and E.v.d.A. provided important intellectual input and final approval of the manuscript.

## Additional information

**Competing interests:** A UK priority patent application (1806118.4) entitled "Macrophage use" was filed on 13th April 2018 by the University of Edinburgh (Inventors: Lesley Forrester and Martha Lopez- Yrigoyen). The pending patent application covers the programming of macrophages with KLF1 activation and the use of cytokines to improve erythroid maturation. The authors declare no other competing interests.

