## [Peer Review File · Nature Communications]

Reviewers' comments:

Reviewer #1 (RBC generation in vitro, culture/expansion/method)(Remarks to the Author):

The authors aimed to make microenvironment (niche) to produce enucleated red blood cells (RBCs) efficiently in the in vitro culture system. Based on the information that macrophages are one of the key players of such niche, the authors tried to produce functional macrophages from iPS cells. They found that an iPS cell line expressing a transcription factor KLF1 (iKLF1.2) could produce better macrophages for this purpose compared to the parent iPS cell line. The results may open a way for in vitro production of enucleated RBCs more efficiently and abundantly.

Major concerns:

#1. The authors showed that the rate of enucleation was improved by their method. However, the results of flow cytometer must be those after viable cells are gated and dead cells are excluded. How about the absolute number? In general, during differentiation process cells are prone to cell death. Was the viability of cells maintained at high levels by their method even after the induction of differentiation? The numbers of analyzed cells in Figure 3B are clearly smaller than those in Figure 3A. Anyway, the authors must show the absolute number of produced enucleated RBCs in comparison between iKLF1.2-derived macrophages and parent iPS-derived macrophages using the same number of CD34+ cells as the starting materials. In addition, the authors must show how many enucleated RBCs can be produced from one CD34+ cell by this method.

#2. In my experiences, the results of staining with CD71 and Hoechst are sometimes difficult to evaluate accurately. For example, double negative cells (CD71-Hoechst- cells) are not necessarily enucleated cells alone. Therefore, the authors must show morphological analysis after collecting double negative cells (CD71-Hoechst- cells) by flow cytometer. The result regarding right of Figure 3B may be appropriate.

#3. If double negative cells (CD71-Hoechst- cells) included non-nucleated cells more or less, the evaluation for rates of enucleation should be performed by morphology after cytopins of cells.

#4. Regrading following description in paragraph 1 of Introduction:

"RBC production from all of these sources is relatively inefficient and a low proportion of the resultant cell populations undergo the enucleation process that marks the final steps of erythroid maturation." I think that relatively efficient production of enucleated RBCs from hematopoietic stem/progenitor cells present in umbilical cord blood was previously reported. The remained problem seemed to be the cost performance. In this context, the method the authors developed this time may be more useful for the production of RBCs from iPS cells or immortalized erythroid cell lines. Therefore, it should be much better to show the results of production of RBCs from iPS cells or immortalized erythroid progenitor cell lines.

#5. If KLF1 is introduced in primary monocytes, can the macrophages which are derived from such monocytes function similarly to those derived from iKLF1.2 cells?

#6. The authors demonstrated only one cell line, iKLF1.2, as a useful iPS cell line to obtain functional macrophages as niche cells. Is it reproducible to generate the useful iPS cell line expressing KLF1 not only from the iPS cell line they used but also from other iPS cell lines? This is very critical when other investigators consider the application of their method.

#7. In the final figure, the authors showed that combination of three humoral factors could support enucleation process. Cell-cell contact is necessary only for phagocytosis of pyrenocytes? Or, cell-cell

contact is also necessary for maturation and enucleation of RBCs? If the absolute numbers of enucleated RBCs produced with and without cell-cell contact are shown, it will become clear.

Minor comments:

#1. With respect to three humoral factors, description is lacking in materials and methods. What company did they purchase those from?

#2. The 1st and 2nd factors shown in Figure 4B (FDCSP and PI16) are commercially available at the moment. Seemingly, they became commercially available during their study. It is better to mention it somewhere.

Reviewer #2 (HSC/iPSC therapy, transplantation, scaled culture)(Remarks to the Author):

The paper by Lopez-Yrigoyen entitled "genetic programming of macrophages generates the first in vitro model for human erythroid island niche" describes a novel approach to generate red blood cells. They show that iPSC-derived macrophages can be genetically programmed to an erythroid island (EI)-like phenotype with KLF1. These EIs were able to generate erythroid cells from cord blood and they further identified KLF1 target genes for potential mediators of macrophage-associated maturation. The studies are well done and the results are presented clearly. The authors nicely show the level of KLF1 expression on iPSC-DMs and the importance of KLF1 for EI-related genes. They identified secreted factors with potential for enhancing maturation and enucleation and showed the effect on cord blood CD34 cells. My main concern with the manuscript is that it does not go beyond the in vitro studies. It is not clear what the efficacy of the erythroid cell production is or would be ie is this clinically meaningful? It is also not clear how functional these red blood cells or erythroid cells are or would be. Without some sort of functional assay to address these questions, I am not convinced that this is a significant enough advance for Nature Communications. Most of the technologies used to generate these findings are well established in the field.

Response to Specific Reviewers' comments:

Reviewer #1 (RBC generation in vitro, culture/expansion/method)(Remarks to the Author):

Major concerns:

#1. The authors showed that the rate of enucleation was improved by their method. However, the results of flow cytometer must be those after viable cells are gated and dead cells are excluded. How about the absolute number? In general, during differentiation process cells are prone to cell death. Was the viability of cells maintained at high levels by their method even after the induction of differentiation? The numbers of analyzed cells in Figure 3B are clearly smaller than those in Figure 3A. Anyway, the authors must show the absolute number of produced enucleated RBCs in comparison between iKLF1.2-derived macrophages and parent iPSC-derived macrophages using the same number of CD34+ cells as the starting materials. In addition, the authors must show how many enucleated RBCs can be produced from one CD34+ cell by this method.

We thank the reviewer for raising this very important point. We have now re-analysed our data and altered the figures to reflect the absolute numbers of enucleated cells that are generated under the different culture conditions. This has actually strengthened our case and very clearly demonstrates that there is a significant increase in the absolute number of enucleated RBCs when the same number of differentiating CD34+ cells are co-cultured with iKLF1.2-derived macrophages compared to parent iPSC-derived macrophages and control cultures (Revised Figure 3C). We have revised Figure 3 and its associated text (page 8, lines 135-141). We refer to a detailed analysis of the calculations in a revised Supplementary Figure S4. These data indicate that contact macrophage co-culture increased the proliferation of erythroid cells and that KLF1-activated macrophages increased this further (Supplementary Figure S4A). Contact macrophage co-culture resulted in a slight increase in viability (Figure S4B) but had no effect on the proportion of erythroid cells in the culture as measured by CD235 expression (Supplementary Figure S5C). However, importantly, the terminal maturation of these CD235a⁺ cells is significantly enhanced by KLF1-activated macrophage co-culture (Supplementary Figure S4D). Taken together, KLF1-activated macrophage co-culture results in a 10-fold increase in the absolute number of mature erythroid cells generated from the same number of CD34+ starting cells (Supplementary Figure S4E).

We also observed an increase in the absolute number of terminally differentiated enucleated erythroid cells in the presence of KLF1-activated macrophages when contact was inhibited (Revised Figure 4A) and when cells were cultured in the presence of the secreted factors compared to control cultures (Revised Figures 4D-F F). Text associated with these figures is revised accordingly (Page 9, lines 159-167: Page 11, lines 206-209)

#2. In my experiences, the results of staining with CD71 and Hoechst are sometimes difficult to evaluate accurately. For example, double negative cells (CD71-Hoechst- cells) are not necessarily enucleated cells alone. Therefore, the authors must show morphological analysis after collecting double negative cells (CD71-Hoechst- cells) by flow cytometer. The result regarding right of Figure 3B may be appropriate.

Thanks for this suggestion, we sorted CD71⁺Hoechst⁻ cells and confirmed that this population does indeed consist of enucleated cells by morphological analysis in cytopspin preparation (Supplementary Figure S3B). Text is amended accordingly (Page 8, lines 130-132).

#3. If double negative cells (CD71-Hoechst- cells) included non-nucleated cells more or less,

the evaluation for rates of enucleation should be performed by morphology after cytopins of cells.

No nucleated cells were apparent in the sorted (CD71-Hoechst- cells) population as demonstrated by cytospin preparations (see above, Supplementary Figure S3B).

#4. Regrading following description in paragraph 1 of Introduction:

“RBC production from all of these sources is relatively inefficient and a low proportion of the resultant cell populations undergo the enucleation process that marks the final steps of erythroid maturation.”

I think that relatively efficient production of enucleated RBCs from hematopoietic stem/progenitor cells present in umbilical cord blood was previously reported. The remained problem seemed to be the cost performance. In this context, the method the authors developed this time may be more useful for the production of RBCs from iPS cells or immortalized erythroid cell lines. Therefore, it should be much better to show the results of production of RBCs from iPS cells or immortalized erythroid progenitor cell lines.

The reviewer is correct in that the enucleation RBCs derived from UCB-CD34⁺ cells is higher than those derived from iPSCs and immortalised progenitors and we have reworded that sentence in our introduction to reflect that point. In this study we used the UCB cell source as the starting point to test the function of the KLF1-activated EI-like macrophages. Using these in co-culture experiments we demonstrate a 10-fold increase in the production of enucleated RBCs and so this could address the cost performance. However, what we believe is the most important point of our study is that this co-culture system provides a powerful experimental system to study factors associated with RBC maturation. Our first “proof of principle” test was to use UCB-derived progenitors but of course, the ultimate aim would be to use that knowledge to improve alternative sources such as iPSC and immortalised erythroid progenitors as the reviewer suggests. We now include data to show the effect of KLF1-activated macrophages on the production of RBCs from iPSCs. Although enucleation efficiency of this source remains lower than UCB-derived cells, we demonstrate a 4-fold increase in the absolute numbers of mature RBCs produced compared to cultures in the absence of macrophages (Page 12, lines 224-238 and new Figure 5).

To emphasise the key point of our study, we have made some changes to the first two paragraphs of our introduction to reflect more accurately the fact that the focus of our paper is the generation of EI-like macrophages and thus a model to study the EI niche and to develop culture strategies for alternative sources (Page 4 lines 17-48).

#5. If KLF1 is introduced in primary monocytes, can the macrophages which are derived from such monocytes function similarly to those derived from iKLF1.2 cells?

It has not been possible to do this experiment because primary monocytes are notoriously difficult to transfect and we do not have a viral construct with the KLF1-ER^{T2} transgene. It is not clear how this experiment would add to our story because it is thought that monocyte have a different developmental origin and may not contribute to the erythroid island niche in vivo.

#6. The authors demonstrated only one cell line, iKLF1.2, as a useful iPS cell line to obtain functional macrophages as niche cells. Is it reproducible to generate the useful iPS cell line expressing KLF1 not only from the iPS cell line they used but also from other iPS cell lines? This is very critical when other investigators consider the application of their method.

We now demonstrate the reproducibility of our approach by showing that additional iPSC lines are able to generate macrophages and their phenotype can be altered by KLF1 activation in a similar manner. We have generated an additional Supplementary

Figure (Supplementary Figure S2) and altered the text accordingly (page 7/8, lines 115-120). It is also important to note that our approach involves targeting the inducible KLF1-ER^{T2} transgene into the AAVS1 locus. We emphasise that point in our discussion and refer to our previous publication that demonstrates the reproducibility of this targeting approach, its use in any iPSC line and that the insertion of transgenes into the AAVS1 locus does not affect macrophage production and function (page 14, lines 273-278).

#7. In the final figure, the authors showed that combination of three humoral factors could support enucleation process. Cell-cell contact is necessary only for phagocytosis of pyrenocytes? Or, cell-cell contact is also necessary for maturation and enucleation of RBCs? If the absolute numbers of enucleated RBCs produced with and without cell-cell contact are shown, it will become clear.

Again, we thank the reviewer for this insightful suggestion. By reanalysing our data based on absolute cell numbers we have been able to understand better the mechanism of action. It is clear that the fold increase in non-contact cultures was not as high as we observed in contact culture suggesting that both cell-cell contact and secreted factors are involved in mediating the effects of KLF1. However, interestingly, when contact was inhibited we observed no effect of KLF1 activation on the proliferation and viability of erythroid cells compared to macrophages in the absence of tamoxifen. This suggests that the phenotypic effect of KLF1 target genes is restricted to the effect on terminal differentiation and enucleation. These data are now presented in Supplementary Figure S6 and described in the text in detail (page 9, lines 159-167).

Minor comments:

#1. With respect to three humoral factors, description is lacking in materials and methods. What company did they purchase those from?

We apologise for this omission and now include that information in the methods section (page 19, lines 392-394)

#2. The 1st and 2nd factors shown in Figure 4B (FDCSP and PI16) are commercially available at the moment. Seemingly, they became commercially available during their study. It is better to mention it somewhere.

Although these factors are commercially available there were no published studies indicating that they had functional activity. We have now clarified that point (Page 11, lines 195-198).

Reviewer #2 (HSC/iPSC therapy, transplantation, scaled culture)(Remarks to the Author):

The paper by Lopez-Yrigoyen entitled “genetic programming of macrophages generates the first in vitro model for human erythroid island niche” describes a novel approach to generate red blood cells. They show that iPSC-derived macrophages can be genetically programmed to an erythroid island (EI)-like phenotype with KLF1. These EIs were able to generate erythroid cells from cord blood and they further identified KLF1 target genes for potential mediators of macrophage-associated maturation.

The studies are well done and the results are presented clearly. The authors nicely show the level of KLF1 expression on iPSC-DMs and the importance of KLF1 for EI-related genes. They identified secreted factors with potential for enhancing maturation and enucleation and showed the effect on cord blood CD34 cells. My main concern with the manuscript is that it does not go beyond the in vitro studies. It is not clear what the efficacy of the erythroid cell production is or would be ie is this clinically meaningful? It is also not clear how functional

these red blood cells or erythroid cells are or would be. Without some sort of functional assay to address these questions, I am not convinced that this is a significant enough advance for Nature Communications.

Efficiency: We now present the data as absolute cell number data) which demonstrates that EI-like macrophages does increase the efficiency of RBC production from UCB by 10-fold (see details in Reviewer, point 1)

Function: With reference to a functional analysis we remind this reviewer that the majority of the experiments to assess the effect of EI macrophages has been performed using UCB CD34+ cells as the source of RBCs. There is ample evidence in the literature that RBCs generated in vitro from UCB are function with respect to their ability to carry oxygen. However, to ensure that the improvements we have made to these protocols do not alter the functionality of the RBCs, we carried out oxygen dissociation experiments (Page 12 lines 240-245. Supplementary Figure S10). We show that there is no significant difference the O2 dissociation curves indicating that the co-cultured erythroid cells are functionally equivalent to UCB-CD34+ derived erythroid cells and adult peripheral blood that was included as a control. To further address this point we performed a more detailed qRT-PCR comparison of cells that are generated +/- macrophage co-culture. We show that co-cultured cells express equivalent globin profiles (indicative of function) and a slight increase in terminal differentiation markers compared to cells cultured under published conditions (Supplementary Figure S5, additional text page 8/9, lines 141-146).

Most of the technologies used to generate these findings are well established in the field.

We would argue that the key point in our paper is that the strategy we have developed to programme macrophages genetically into a specific phenotype is a novel approach. This would be applicable to many biological systems and clinical applications where macrophages play a key role and thus is likely to be of interest to a broad audience. Thus we have reworded the first two paragraphs of our introduction to emphasize this point.

REVIEWERS' COMMENTS:

Reviewer #1 (Remarks to the Author):

The authors well addressed my concerns. This reviewer sincerely hopes further progresses of the relevant fields.

Reviewer #2 (Remarks to the Author):

The authors have addressed my concerns.